# N-TiO₂-Coated SiC Foam for the Treatment of Dyeing Wastewater under Blue Light LED Irradiation

Wen Sun, Xuankun Li *, Jingtao Zou and Han Guo

College of Chemistry and Chemical Engineering, Shanghai University of Engineering Science, Shanghai 201620, China; kame281@163.com (W.S.); zoujingtao2022@163.com (J.Z.); guohan.work@sues.edu.cn (H.G.)
* Correspondence: li.xuankun@sues.edu.cn

**Abstract:** TiO₂ is widely used for the photocatalytic degradation of organic pollutants in wastewater, but the practical applications of the photocatalyst are limited due to its poor visible light absorption and low recovery rate. In this study, high production of nitrogen-doped TiO₂ was achieved by a hydrolysis precipitation method; the obtained N-TiO₂ had a small crystallite size and good dispersibility. The effect of calcining temperature on the photocatalytic performance of N-TiO₂ was evaluated through the degradation of methylene blue (MB) in a blue light LED irradiation cylinder, it was found the N-TiO₂ calcined at 400 °C showed the best photocatalytic activity. Then the N-TiO₂ was immobilized on SiC ceramic foam by dip-coating with PVA as the binder. The prepared N-TiO₂/SiC foam showed excellent photocatalytic activity under blue light LED irradiation; as high as 96.3% of MB was degraded at optimum conditions. After five cycles of MB photodegradation, the photocatalytic activity of N-TiO₂/SiC foam only changed slightly, which makes it a promising photocatalytic material for wastewater treatment.

**Keywords:** nitrogen-doped TiO₂; SiC foam; LED irradiation; photodegradation; methylene blue





## 1. Introduction

Textile industrial dyeing wastewater not only affects the transparency of receiving waters, but also arouses concerns about its biological toxicity, carcinogenicity, and teratogenicity on aquatic organisms [1]. It has been reported that about 10 to 15% of the dyes used in this industry are discharged into the wastewater; therefore, it is urgent to treat the dyeing wastewater before discharge [2]. However, dyeing wastewater is less biodegradable compared to common wastewater, and consequently, more and more researchers are relying on advanced oxidation processes (AOPs) to resolve the problem [3]. Radicals generated in AOPs, such as hydroxyl radicals (·OH), can effectively decompose organic pollutants into harmless inorganic carbon dioxide and water. For example, the Fenton process introduces $Fe^{2+}$ into an $H_2O_2$ solution to generate abundant ·OH, and it has been applied for the degradation of plenty of organic pollutants [4]. However, the loss of iron ions and the formation of solid sludge requires further treatment. In addition to the high cost of chemicals, $H_2O_2$ also limits the application of Fenton-based AOPs [5]. Sulfate radical advanced oxidation processes (SR-AOPs) are also efficient solutions for the treatment of organic pollutants. In these processes, peroxymonosulfate (PMS) and peroxydisulfate (PDS) are activated to generate highly oxidative sulfate radicals [6], but the production of a high concentration of $SO_4^{2-}$ may introduce new problems for water treatment [7].

Heterogeneous photocatalytic oxidation is a cleaner AOP technology for the degradation of organic pollutants. In brief, this process employs semiconductors as catalysts. When excited by light of a certain wavelength, reactive oxygen species (ROS), such as hydroxyl radicals (·OH) and superoxygen radicals ($O_2^-$), could be generated to mineralize the organic pollutants [8]. Among all the photocatalysts, TiO₂ is the most widely applied in both research and industry due to its low cost, high chemical stability, and non-toxic

properties [9]. Commercial $TiO_2$ photocatalysts, such as Degussa P25 and Hombikat UV100 have already been applied in VOC degradation and water treatment [10]. In the solar energy reaching the Earth's surface, UV light comprises only 4%, while visible light comprises approximately 43% [10]. However, because of a wide band-gap and fast electron–hole pair recombination of $TiO_2$ semiconductors, its visible-light photocatalytic activity is less effective [11]. To overcome these drawbacks, doping $TiO_2$ with non-metals (e.g., C, N, and B) or transition metals (e.g., Co, Cu, Ce, and Mn) have been proved to be effective [10]. Among all the doping elements, nitrogen has an atomic size and electronic characteristics that are similar to oxygen. Yu et al. applied $NH_4Cl$ as the nitrogen source in a modified sol-gel process. The produced N-doped $TiO_2$ was characterized to have a small particle size and high photocatalytic activity under visible light [12]. Jin et al. synthesized a $N-TiO_2$ of a narrow band-gap (2.92 eV) by the hydrothermal method, which degraded 99.53% of Norfloxacin within a mere 30 min under visible light irradiation [13]. $N-TiO_2$ can be synthesized through various methods, such as the sol-gel process [12], the hydrothermal method [13], the mechanochemical reaction [14], and so on. Among all the synthesis methods, the hydrothermal method was reported to be simple in operation, short in reaction time, and high in yield.

Considering that the recovery and reuse of $TiO_2$ powder catalysts is difficult, it is necessary to anchor the $TiO_2$ to a supporting material. The most widely used supports include glass beads [15], zeolites [16], porous silica particles [17], stainless steel [18], and so on. SiC foam is a 3D support material with high chemical and thermal stability, which ensures a large contact surface between the supported photocatalyst and the effluent. Moreover, the macroporosity of 3D-SiC foam also provides a large internal surface, which provides conditions for supporting photocatalysts [19]. In addition, it has been reported that the heterostructures of $TiO_2$ with other semiconductors can effectively reduce the recombination rate of electron–hole pairs and improve the photocatalytic reaction efficiency [20–22].

In this study, the hydrolysis precipitation method was applied to improve the yielding rate of $N-TiO_2$. Then, the synthesized $N-TiO_2$ was immobilized on SiC ceramic foam by dip-coating with PVA as the binder. The photocatalytic performance of the obtained $N-TiO_2$/SiC foam photocatalyst was evaluated through the degradation of methylene blue, a typical azo-dye in textile wastewater. To lower the energy consumption of the photocatalytic process, a blue light LED cylinder was designed to irradiate the catalytic system. The effects of the initial dye concentration, pH, $N-TiO_2$ loading rate, etc., on the photodegradation of the MB, as well as the reusability of the foam catalyst were investigated. The aim of this study was to develop a reusable foam catalyst and reveal the optimal experimental conditions for the heterogeneous photocatalytic degradation of MB.

## 2. Materials and Methods

### 2.1. Chemicals and Materials

Titanium (IV) butoxide (TBT, 99% purity) and ammonium hydroxide (25–28%) were obtained from damas-beta. Polyvinyl alcohol (PVA) and methylene blue were provided by Aladdin and SCR (Shanghai, China), respectively. SiC ceramic foams (20 ppi) were purchased from Defu Newmaterial (Shandong, China). Each foam had a 5 cm diameter and a weight of around 11.0 g. The received SiC foams were firstly ultrasonically cleaned with DI water and then calcinated to eliminate residual organic carbon.

### 2.2. Synthesis of $N-TiO_2$

The TBT (10 mL) was added dropwise into a 5% $NH_4OH$ (100 mL) solution in an ice bath. A white precipitate gradually formed under vigorous stirring of the solution. After 20 min, the precipitate was separated and washed with deionized (DI) water. After being dried at 80 °C for 1 h, the powder was further calcinated at various temperatures ranging from 300 to 500 °C for 2 h (heating rate of 5 °C/min) [23].

### 2.3. Immobilization of N-TiO₂/SiC Foams

Firstly, the PVA (1.5 g) was dissolved in deionized water (60 mL, 80 °C) and then the N-TiO₂ (12 g) was added. Next, the mixture was transferred into a planetary ball mill (ZQM-P2, MITR, China) and ball-milled for 90 min. After filtering through a 100-mesh filter, the stable N-TiO₂ slurry was used for the coating of SiC foams. Briefly, SiC foams were immersed in the N-TiO₂ slurry and dried at room temperature for 60 min; this process was conducted twice. After drying at 80 °C overnight, the coated foams were further calcinated at 400 °C for 2 h with a heating rate of 5 °C/min. The SiC foams were subjected to different runs of a coating procedure to achieve the expected N-TiO₂ loading rate [19].

### 2.4. Characterizations

The phase structures of the samples were recognized by X-ray diffraction (XRD, X Perp PRO, PANalytical B.V, The Netherlands) with Cu Kα radiation over a 2θ range of 20°–80°. The microstructures of the surfaces and cross-sections of the SiC foams coated with N-TiO₂ were examined using field emission scanning electron microscopy (SEM, SU8010, HITACHI, Tokyo, Japan). A UV-visible spectrophotometer (UV-vis, UV-2600, SHIMADZU, Shimadzu, Japan) was utilized to test the optical properties.

### 2.5. Experimental Setups

The photocatalytic degradation of MB was carried out in a laboratory pilot scale photoreactor in the recirculation mode. As shown in Figure 1, two pieces of N-TiO₂/SiC foam were placed in an acrylic column (f = 55 mm, h = 280 mm), and the column reactor was surrounded by a customized 40 W blue light LED cylinder (irradiation wavelength of 465 nm). The MB solution (500 mL) was recirculated through the photoreactor using a peristaltic pump. Before irradiation, the solution was recirculated for 30 min in the dark to let the MB reach adsorption–desorption equilibrium in the foam surface. During the photocatalytic degradation, 3 mL aliquots were taken at certain time intervals and measured by the UV-vis (λ = 664 nm) to determine the concentration of MB.

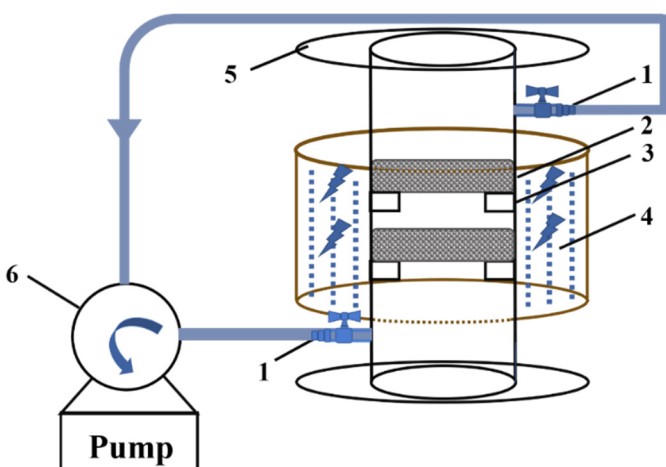

**Figure 1.** Schematic diagram of the photo-reactor: (1) inlet and outlet, (2) N-TiO₂/SiC foams, (3) catalyst holder, (4) LED lights, (5) acrylic tube, and (6) peristaltic pump.

## 3. Results

### 3.1. Catalyst Characterization

The effects of different calcination temperatures on the crystal structures and phases of N-TiO₂ particles were investigated by XRD. The catalysts prepared at different calcination temperatures only had an anatase phase. Figure 2 shows the XRD patterns of the synthesized N-TiO₂. It can be observed that the peaks are in accordance with the anatase phase TiO₂ (PDF#21-1272) with a tetragonal crystal structure. The obtained peaks at the 2θ values of 25.3°, 36.9°, 37.8°, 38.6°, 48.0°, 53.9°, 55.1°, 62.7°, 68.8°, 70.3°, and 75.0° were

indexed to the (101), (103), (004), (112), (200), (105), (211), (204), (116), (220), and (215) planes respectively. The catalyst remaining in an anatase phase might be due to the fact that the N partially replaced the lattice ions, which hindered the phase transformation of the TiO$_2$ from anatase to rutile. Meanwhile, it is worth noting that the diffraction peaks of the N-TiO$_2$ calcined above 400 °C were intense and sharp, which proved the catalyst is synthesized with high crystallinity. However, N-O or Ti-N peaks were not detected; this may be due to the fact that N was introduced at very low levels during the synthesis [24].

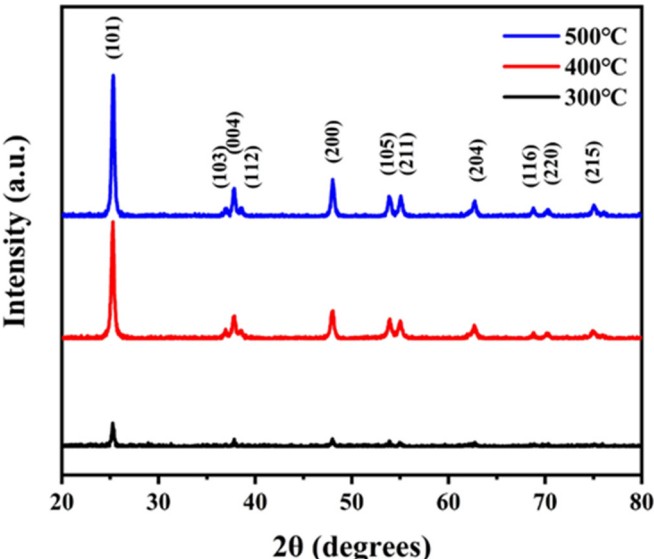

**Figure 2.** The XRD patterns of N-TiO$_2$ photocatalysts with different calcination temperatures.

The average crystallite sizes of the prepared samples were judged by the Scherrer formula ((Equation (1)).

$$D = \frac{k\lambda}{\beta cos\theta} \tag{1}$$

where $D$ is the average thickness of the crystal grain perpendicular to the crystal plane (nm), $\lambda$ is the wavelength of the X-ray irradiation (0.154056 nm), $k$ is the Scherrer constant (0.89), $\beta$ is the half-height width of the diffraction peak of the measured sample, which needs to be converted into radians (rad) during the calculation process, and $\theta$ is the brag diffraction angle (degrees).

The crystallite sizes of N-TiO$_2$-300, N-TiO$_2$-400, and N-TiO$_2$-500 were calculated as 19.53, 21.07 and 22.71 nm, respectively, indicating that the crystallite sizes of the as-prepared photocatalysts increased with the increase in calcination temperature.

The optical response of the N-TiO$_2$ was analyzed through a UV-visible spectrometer, and the scanning range of the UV-vis spectrum was from 300 to 800 nm. The Kubelka–Munk model (Eg = 1239.8/$\lambda$) was applied to calculate the band-gap energy of N-TiO$_2$ [25]. It is well known that the band-gap of pure anatase phase TiO$_2$ is around 3.2 eV. As portrayed in Figure 3b, a remarkable red shift of the absorbance peak is observed for N-TiO$_2$-400. The N-TiO$_2$ with the lowest band-gap energy of 2.47 eV was obtained under 400 °C calcination. This may have been due to the N atoms being doped into the TiO$_2$ by replacing the lattice oxygen during the hydrolysis procedure. The red shift of the absorbance peak of N-TiO$_2$-400 indicates that the catalyst is likely to be excited under visible blue light irradiation.

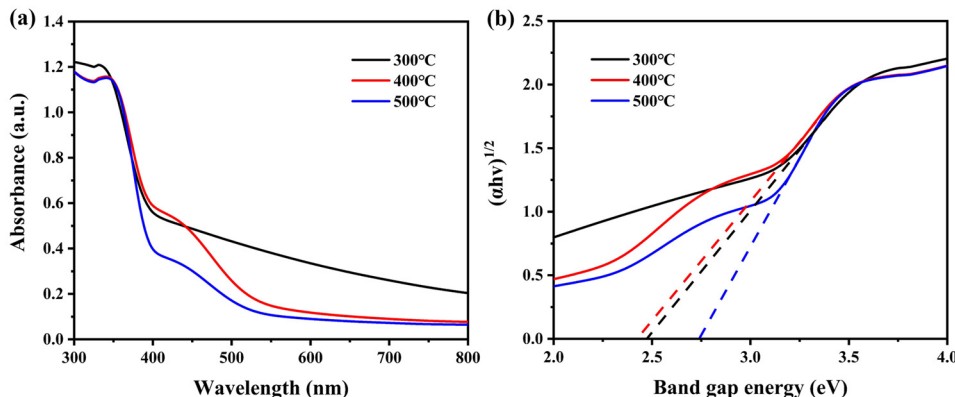

**Figure 3.** (**a**) UV-visible absorption spectra of N-TiO$_2$ prepared under three different calcination temperatures. (**b**) The plot of the transformed Kubelka–Munk function versus the wavelength of light.

As shown in Figure 4a–c, the calcination temperature has a significant effect on the morphology of N-TiO$_2$. The average particle sizes of N-TiO$_2$-300, N-TiO$_2$-400, and N-TiO$_2$-500 were 400, 500 and 800 nm, respectively, and the size of N-TiO$_2$-400 had good uniform dispersion. It can be concluded that with the decrease in the calcination temperature, the particle size decreased and the specific surface area increased, which was beneficial to the adsorption of reactants and accelerated the photocatalytic reaction rate. After dipping the SiC foam in N-TiO$_2$ suspension and annealing it at 400 °C, the surface morphology of the N-TiO$_2$/SiC foam was observed by SEM. As shown in Figure 4d–f, the surface of the SiC foam support was uniformly covered by a N-TiO$_2$ layer that had a thickness of 49 μm. This shows that PVA acting as a binder can significantly improve the deposition and stability of N-TiO$_2$ particles in SiC foam.

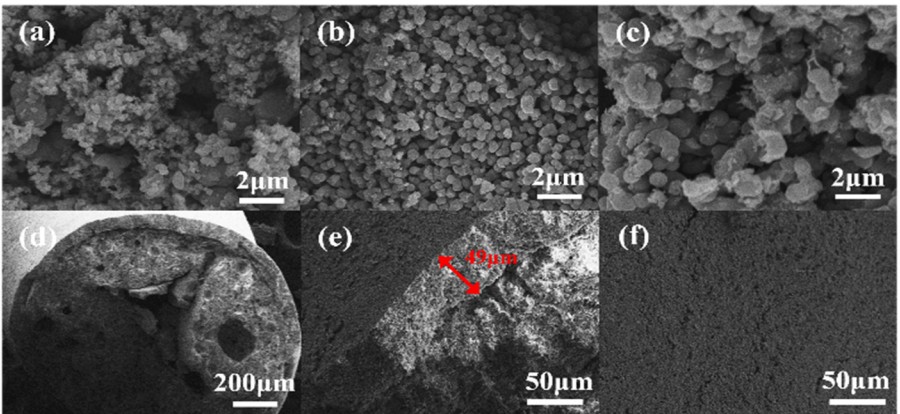

**Figure 4.** SEM images of N-TiO$_2$ prepared under different calcination temperatures and N-TiO$_2$/SiC foam: (**a**) N-TiO$_2$-300, (**b**) N-TiO$_2$-400, (**c**) N-TiO$_2$-500, (**d,e**) cross-section, and (**f**) outer surface.

*3.2. Photocatalytic Degradation of MB by N-TiO$_2$*

Before immobilization, the photocatalytic activities of N-TiO$_2$ calcined at different temperatures were verified. Briefly, 100 mg photocatalyst was dispersed into 100 mL MB (15 mg L$^{-1}$) solution in a beaker and irradiated by a 40 W blue light LED cylinder ($\lambda$ = 465 nm). As shown in Figure 5, before irradiation, 28.8%, 26.5%, and 23.0% of the MB was adsorbed on N-TiO$_2$ calcined at 300, 400 and 500 °C, respectively. The reason why N-TiO$_2$ calcined at 300 °C exerted the highest adsorption capability may be due to the presence of amorphous carbon. With irradiation under blue light, N-TiO$_2$-400 showed the best photocatalytic degradation capability for MB. This result is consistent with the previous characterization of N-TiO$_2$ calcined under different temperatures. The incorporation of N into the crystal lattice of TiO$_2$ narrowed its band-gap and improved the catalytic efficiency

under a wavelength of 465 nm. N-TiO$_2$ calcined at 400 °C had the best photocatalytic performance, and we will take this sample as an example for systematic research.

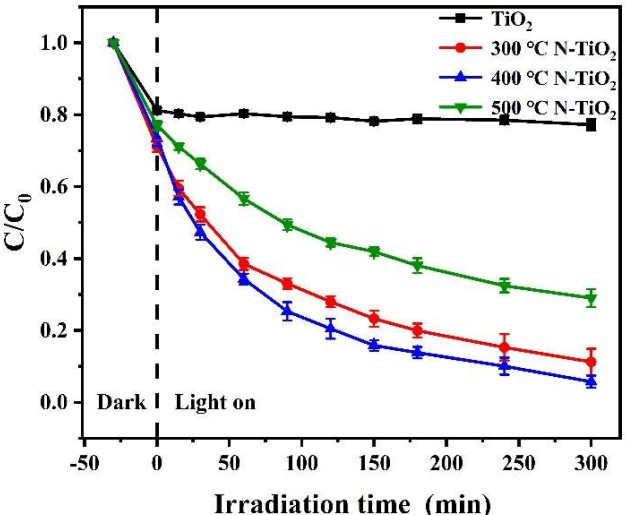

**Figure 5.** Photocatalytic degradation of MB by TiO$_2$ and N-TiO$_2$ calcinated under different temperatures.

### 3.3. N-TiO$_2$/SiC Foam Photocatalytic Activity

### 3.3.1. Effect of Initial MB Concentration

In order to evaluate the photocatalytic treatment potential of MB by N-TiO$_2$/SiC foam, the effect of the MB concentration (10, 15 and 20 mg/L) on degradation efficiency was studied under the conditions of a N-TiO$_2$ loading of 17 ± 1 wt% and pH around 7.0. As shown in Figure 6, the degradation efficiency reached 89.0% when the MB concentration was 10 mg/L, and it remained at almost the same value as the MB concentration further increased to 15 mg/L. However, as the MB concentration reached 20 mg/L, its degradation efficiency remarkably declined to 77.5%. This is because, with the increase in MB concentrations, the pollutant molecules competed for the limited reactive sites on the surface of N-TiO$_2$/SiC foam. In addition, the MB was a blue dye that could significantly reflect blue light irradiation, and a high concentration of MB could significantly inhibit the penetration of photons in the solution, which resulted in the decrease in photo-induced radicals on the catalyst surface [26].

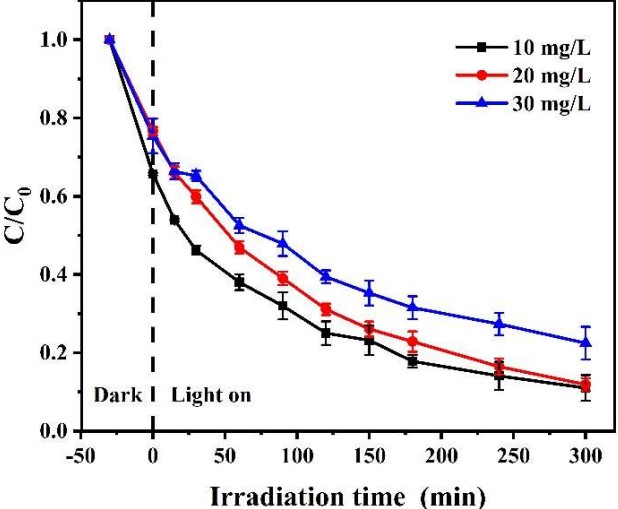

**Figure 6.** Effect of initial MB concentration on photocatalytic degradation efficiency.

### 3.3.2. Effect of Initial pH

Figure 7 shows the effect of the initial solution pH on the photocatalytic degradation of 15 mg/L MB with $17 \pm 1$ wt.% $N\text{-}TiO_2/SiC$ foam as a catalyst under blue light irradiation. It was observed that with the increase in pH, the photocatalytic degradation efficiency of the MB also increased. When the pH value was adjusted to 11, the degradation efficiency of the MB reached as high as 96.3%. Generally, the photocatalytic degradation efficiency of the MB was significantly higher in the alkaline solution than in the acidic condition. Since MB is a cationic dye, it was positive in charge when dissolved. In alkaline solution, $N\text{-}TiO_2$ was reported to be negatively charged [27]. Therefore, due to the attraction between the opposite charges, the adsorption of MB on the catalyst surface was improved, which further facilitated the photodegradation efficiency.

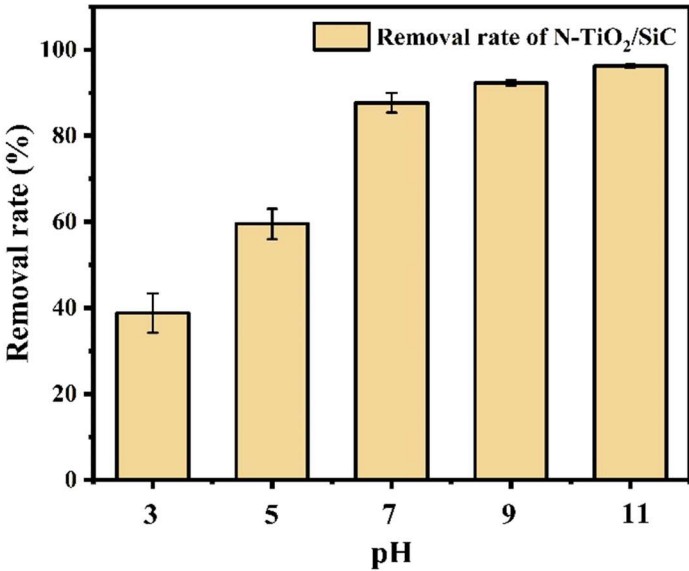

**Figure 7.** Effect of initial solution pH on the photocatalytic degradation efficiency.

### 3.3.3. Effect of Catalyst Loading

To improve the MB photodegradation and lower the cost of the $N\text{-}TiO_2/SiC$ foam photocatalyst, the effect of the $N\text{-}TiO_2$ loading amount on SiC foams was studied under the conditions that the MB concentration was 15 mg/L and the pH was about 7. The SiC foams were coated from 1 to 3 times according to the procedure described in Section 2.3, and the resulting $N\text{-}TiO_2$ loading ratio on the SiC foams ranged from $7 \pm 1$ wt.% to $28 \pm 1$ wt.%. Next, the photocatalytic activity of the prepared $N\text{-}TiO_2/SiC$ foams was evaluated according to the procedure described in Section 2.4. The photocatalytic reaction efficiency was also evaluated by the pseudo-first-order kinetic equation, as follows:

$$-\ln\left(\frac{C}{C_t}\right) = kt \tag{2}$$

where $C$ is the residual concentration of MB at time $t$, $C_t$ is the concentration of MB after dark reaction equilibrium, $k$ is the apparent reaction rate constant, and $t$ is the reaction time [28].

As shown in Figures 8 and 9, the MB removal efficiency improved from 68.0% to 88.1% as the $N\text{-}TiO_2$ loading ratio increased from $7 \pm 1$ wt.% to $17 \pm 1$ wt.%. Meanwhile, when the $N\text{-}TiO_2$ loading ratio was $17 \pm 1$ wt.%, the apparent reaction rate constant also reached a maximum of 0.0066 $\text{min}^{-1}$. The increase in the $N\text{-}TiO_2$ immobilization provided more active sites that could not only adsorb more organic pollutants but also generate oxidative radicals. It is worth noting that bare SiC foams also showed little photocatalytic activity under blue light LED irradiation. According to past studies, this is because SiC is also

a semiconductor, and its band-gap varies from 2.4 to 3.4 eV, which, to some extent, can degrade MB under visible light. Moreover, the synergistic effect between the $TiO_2$ particles and the SiC support was found to be able to reduce the electron–hole recombination [29], thus increasing the photocatalytic activity of $N-TiO_2/SiC$ foams.

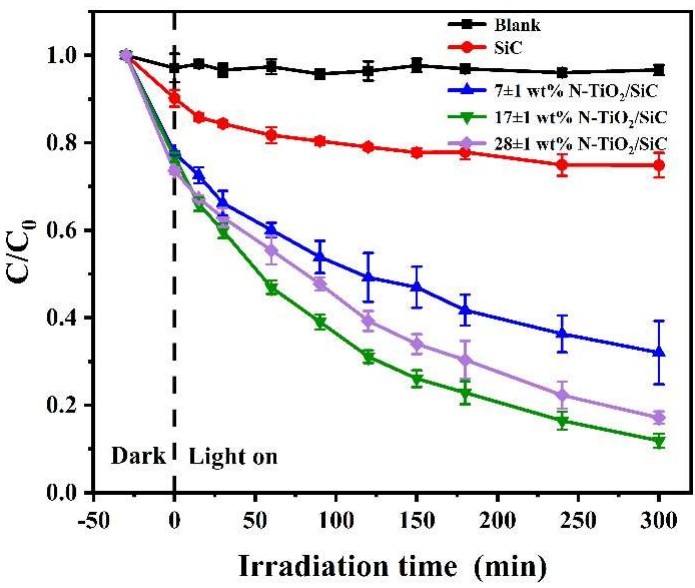

**Figure 8.** Effect of catalyst loading on MB photodegradation.

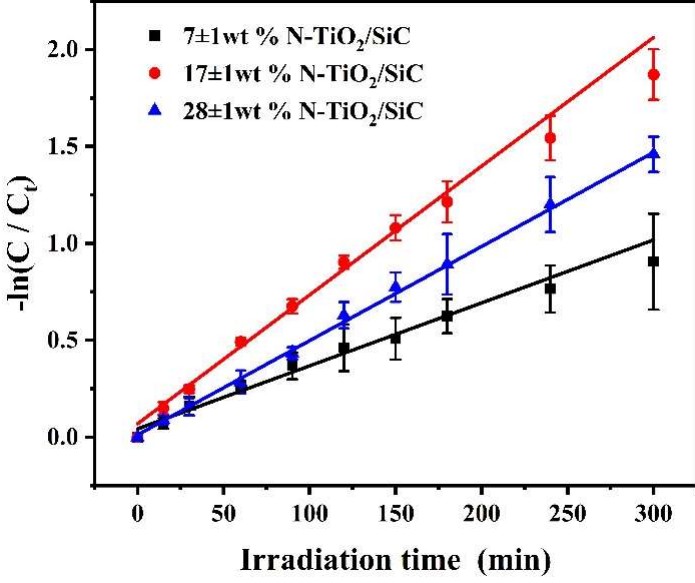

**Figure 9.** Effect of catalyst loading on MB photodegradation kinetics.

However, as the $N-TiO_2$ loading ratio further increased to $28 \pm 1$ wt.%, the MB removal rate reduced to 82.9%, and the apparent reaction rate constant also decreased to 0.0046 $min^{-1}$. The decline of the MB removal rate may have been due to excessive $N-TiO_2$ particles clogging the pores of the SiC foams, resulting in the decrease in the surface area and number of reacting sites. Considering both the photodegradation efficiency and the cost of photocatalysts, a double coating of the foam ($N-TiO_2$ $17 \pm 1$ wt.%) was selected as the optimal catalyst immobilization procedure for further research.

### 3.4. Stability of N-TiO$_2$/SiC Foam

The stability and recyclability of a photocatalyst are among the top concerns in developing a commercial catalyst. The stability of photocatalytic activities and the reusability of N-TiO$_2$/SiC foams were investigated by five consecutive photocatalytic degradations of MB. As shown in Figure 10, the photocatalytic performance of N-TiO$_2$/SiC foams after five successive cycles still remained stable. In the fifth reuse, the final photodegradation efficiency of MB reached 82.3%, which was only 7.4% lower than that achieved in the first-time application. The decrease in the photodegradation efficiency of MB may have been due to the slight weight loss of the catalyst during the recycling process. Therefore, it can be concluded that the novel N-TiO$_2$/SiC foam has considerably high stability when it is applied to the photocatalytic degradations of organic pollutants.

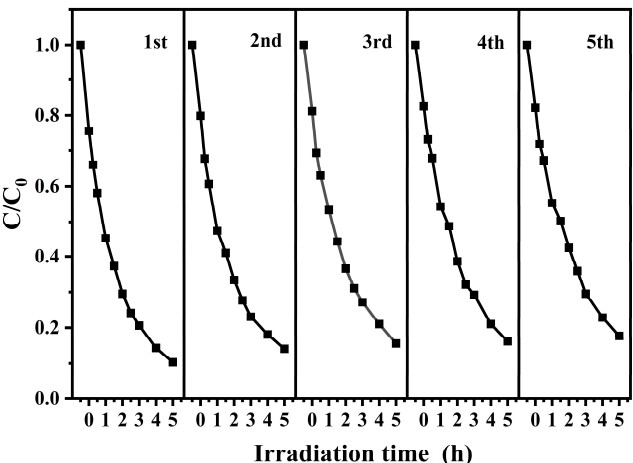

**Figure 10.** Photocatalytic activities of N-TiO$_2$/SiC foam in five cycles of MB treatment.

## 4. Conclusions

In this study, a hydrolysis precipitation method was applied to improve the yielding rate of N-TiO$_2$. The effect of the calcining temperature on the photocatalytic performance of N-TiO$_2$ was evaluated through the degradation of MB in a blue light LED irradiation cylinder. It was found that the N-TiO$_2$ calcined at 400 °C showed the best photocatalytic activity. Then, N-TiO$_2$ powder was immobilized on SiC ceramic foam through dip-coating with PVA as the binder. It was observed that the N-TiO$_2$ was uniformly coated on the SiC foam. The prepared N-TiO$_2$/SiC foam showed excellent photocatalytic activity under blue light LED irradiation; as high as 96.3% of MB can be degraded by a double dip-coated catalytic foam (N-TiO$_2$ 17 ± 1 wt.%) under optimum conditions. The prepared N-TiO$_2$/SiC foam showed significantly higher photocatalytic activity in neutral or alkaline solutions than in acidic conditions. After five cycles of MB photodegradation, the photocatalytic efficiency of the N-TiO$_2$/SiC foam only dropped by 7.4%, which makes it a promising photocatalytic material for wastewater treatment.

**Author Contributions:** Conceptualization, W.S. and X.L.; methodology, W.S. and X.L.; formal analysis, W.S., J.Z. and H.G.; data curation, W.S.; writing—original draft preparation, W.S.; writing—review and editing, X.L. All authors have read and agreed to the published version of the manuscript.

**Funding:** This research was funded by Shandong Natural Science Foundation grant number (ZR2021 QE223).

**Institutional Review Board Statement:** Not applicable.

**Informed Consent Statement:** Not applicable.

**Data Availability Statement:** Not applicable.

**Conflicts of Interest:** The authors declare no conflict of interest.

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
