# Peer review of "N-TiO2-Coated SiC Foam for the Treatment of Dyeing Wastewater under Blue Light LED Irradiation"

_coatings, doi:10.3390/coatings12050585_

Round 1
Reviewer 1 Report
- The author should highlight the novelty of the manuscript explicitly
- The authors are claimed the 5 times reusability of TiO2/SiC hybrids. I was wondering how only dip coating is so effective to bound TiO2 with SiC. Please elaborate it.
- The authors need to highlight the importance of heterostructures creation for photocatalysis for that authors should read and cite Fabrication of ZnO–TiO2 nanohybrids for rapid sunlight driven photodegradation of textile dyes and antibiotic residue molecules." Optical Materials107 (2020): 110138; Two-dimensional MoS 2 nanosheet-modified oxygen defect-rich TiO 2 nanoparticles for light emission and photocatalytic applications." New Journal of Chemistry 44, no. 35 (2020): 14936-14946; Enhanced sunlight driven photocatalytic activity of In2S3 nanosheets functionalized MoS2 nanoflowers heterostructures." Scientific reports 11, no. 1 (2021): 1-14.
- The authors should provide the XRD spectrum of pristine TiO2 and N doped TiO2.
- The authors should provide the UV-DRS and Tauc plots studies of TiO2/SiC.
- PL studies hould be provided for TiO2 and TiO2/SiC.
- The photocatalysis scheme for TiO2/SiC systme should be provided.
Reviewer 2 Report
In this work, Ni-TiO2 photocatalyst was prepared and coated on SiC foam for the photocatalytic degradation of methylene blue dye. Characterization by XRD, UV-vis DRS, and SEM analyses have been performed to provide information on the physicochemical properties of the photocatalyst. This work is interesting and suitable for the scope of the journal and broad audiences. However, some parts of the manuscript should be modified and/or explained. Therefore, I suggest acceptance after the following issues have been addressed:
Specific comments:
- Page 2, line 45: “super oxygen radicals” should be “superoxide radicals”.
- Reference method for the synthesis of N-TiO2 and N-TiO2 coating should be provided. It was not clearly stated in the methodology which part of the synthesis was the hydrothermal method used? Why was PVA selected as the binder?
- The XRD spectrum of TiO2 only should also be included for comparison. Similarly, the results for the photocatalytic degradation of MB by TiO2 only should be included in Figure 5.
- Page 5, line 171: “Figure 5” should be “Figure 4”.
- The surface charge of N-TiO2 should be determined to support the discussion in Section 3.3.2.
- Provide a reference for equation 2.
- The results of photolysis should also be included in Figure 8 for comparison.
- The authors should include some characterizations, such as SEM, XRD, and TGA, to study the stability of the photocatalyst after reusability.
- Error bars must be added in all figures related to experimental parts.
- Technical comments:
- Some minor typos and grammatical errors were still present in this paper. Therefore, I would urge the authors to proofread their manuscript again before publication.
- Do not start a sentence with a numeral. Please revise the sentences.
Reviewer 3 Report
The paper presents very interesting and important subject concerning the application of N-TiO2 for the wastewater treatment. The research presented is very interesting and valuable. The paper is written properly and good containing the main points necessary for scientific papers. The abstract presents the most important aspects of this work. My one remark before the final acceptance is that the summary and conclusion part should be written in points. The results of work will be more readable for readers, therefore.
Reviewer 4 Report
The authors provide a study concerning the photo-degradation of methylene blue on a new type of catalyst made of N-TiO2 supported on SiC foam. Their study is systematic and the investigations are relatively well described. Nevertheless before being accepted for publication it would be reasonable to clarify some experimental conditions :
1 in sections 3.3.1. what was the amount of catalyst and the pH value
2 in section 3.3.2 what was the amount of catalyst
3 in section 3.3.3 what was the concentration of MB and the pH value
From a practical point of view is should be better to consider the photo-degradation rate reached at pH 8.5-9 even if the values obtained at pH 11 are higher, since the admissible pH for water discharge in the effluents is in the range of 6.5-8.5.
It would also be interesting to check if the photo-degradation was total or there are still traces of non-colored organic compounds in the treated water. To determine this, the determination of COD or TOC would be necessary.
Besides the above mentioned aspects, the authors should also revise the English spelling all along the manuscript.
Round 2
Reviewer 1 Report
Now the quality of the manuscript is improved and can be accepted in its present form.
Reviewer 2 Report
The paper has been revised satisfactorily and can be accepted in its current form.